# Fabrication, Facilitating Gas Permeability, and Molecular Simulations of Porous Hypercrosslinked Polymers Embedding 6FDA-Based Polyimide Mixed-Matrix Membranes

**DOI:** 10.3390/molecules28052028

**Published:** 2023-02-21

**Authors:** Chaohua Song, Longfei Peng, Yinhui Li, Yawei Du, Zan Chen, Weixin Li, Cuijia Duan, Biao Yuan, Shuo Yan, Sibudjing Kawi

**Affiliations:** 1School of Chemical Engineering and Technology, Hebei University of Technology, Tianjin 300130, China; 2Department of Chemical and Biomolecular Engineering, National University of Singpore, 4 Engineering Drive 4, Singapore 117585, Singapore; 3Key Laboratory of Membrane and Membrane Process, China National Offshore Oil Corporation Tianjin Chemical Research & Design Institute, Tianjin 300131, China

**Keywords:** 6FDA-based polyimide, hypercrosslinking polymers, gas transport, molecular simulations

## Abstract

Novel polymers applied in economic membrane technologies are a perennial hot topic in the fields of natural gas purification and O_2_ enrichment. Herein, novel hypercrosslinked polymers (HCPs) incorporating 6FDA-based polyimide (PI) MMMs were prepared via a casting method for enhancing transport of different gases (CO_2_, CH_4_, O_2_, and N_2_). Intact HCPs/PI MMMs could be obtained due to good compatibility between the HCPs and PI. Pure gas permeation experiments showed that compared with pure PI film, the addition of HCPs effectively promotes gas transport, increases gas permeability, and maintains ideal selectivity. The permeabilities of HCPs/PI MMMs toward CO_2_ and O_2_ were as high as 105.85 Barrer and 24.03 Barrer, respectively, and the ideal selectivities of CO_2_/CH_4_ and O_2_/N_2_ were 15.67 and 3.00, respectively. Molecular simulations further verified that adding HCPs was beneficial to gas transport. Thus, HCPs have potential utility in fabrication of MMMs for facilitating gas transport in the fields of natural gas purification and O_2_ enrichment.

## 1. Introduction

Gas separation is an essential separation process in natural gas purification, flue gas separation, and O_2_ enrichment [1,2,3,4,5,6]. Membrane technologies are usually used for gas separation thanks to their environmentally friendly processes and low energy consumption [7,8,9]. Unfortunately, permeability has always been a problem restricting gas separation performance in membrane technology [10]. In general, gas separation membranes are divided into homogeneous and heterogeneous structures. Dense membranes with homogeneous structures have poor permeation and thus are restricted in industrial applications.

Mixed-matrix membranes (MMMs) with heterogeneous structures are excellent candidates for gas separation because they can facilitate gas transport with the help of fillers with porous structures and functional groups [11]. Porous structures promote gas transport and functional groups and are beneficial to gas dissolution–diffusion, which breaks through low permeation and realizes highly effective gas separation [12,13,14].

MMMs consist of fillers with porous structures in a polymeric matrix, including rubber polymers and glassy polymers. Normally, glassy polymers such as polysulfone [15,16], cellulose [17,18], and polyimide [19,20,21] can separate gas well. Of these, polyimide (PI), with its highly thermal stability, good chemical stability, and strong mechanical stability, has been extensively used as a gas separation membrane [22,23]. In addition, the large fractional free volume (FFV) of aromatically fluorinated PI is beneficial to gas transport [24,25]. However, a polymer matrix with a large FFV will result in gas selectivity reduction that falls into conventional trade-off effects. Thus, preparing PI MMMs with high permeability and suitable selectivity is still an important challenge.

In recent years, researchers have focused on using fillers with tailored porous structures and functional groups to improve gas transport and overcome the trade-off effect. For example, Koros et al. synthesized a zeolite-like MOF (ZMOF) filler for preparation of ZMOF/PI MMMs. Gas permeation experiments have shown that ZMOF/PI MMMs enhance CO_2_ permeability due to the incorporation of ZMOF fillers with affinity CO_2_ frameworks, which is beneficial to enhancing CO_2_ sorption and facilitating gas diffusivity [26]. Yeong et al. prepared zeolite T as an inorganic filler for the fabrication of zeolite T/PI MMMs in CO_2_/CH_4_ separation. Zeolite T could effectively improve CO_2_ and CH_4_ transport properties. Compared to pure PI film, the permeability of zeolite T/PI MMMs toward CO_2_ increased by 80%. Additionally, zeolite T/PI MMMs also improved CO_2_ plasticization resistance [27]. Chung et al. fabricated zeolitic imidazolate framework (ZIF)-71 inorganic fillers with diameters of less than 100 nm, and these fillers were compatibly incorporated in PI membranes. ZIF-71 in the PI membranes enhanced gas permeation, and the permeability of ZIF-71/PI MMMs toward pure CO_2_ was threefold higher than that of a pure PI membrane [28]. In short, developing a perfect filler using MMMs is an effective approach to promoting gas transport and breaking through the trade-off effect.

To date, different fillers, such as silica nanoparticles [29,30], COFs [31], MOFs [32,33], ZIFs [34,35], and carbon nanotubes [36,37] have been used to prepare polyimide MMMs and enhance permeability and selectivity. The permeability and selectivity of the MMMs were greatly improved after incorporation of the fillers. However, incompatible interfaces between fillers and polymer matrices have always been a problem due to the effects of defects, caused by weak compatibility, on gas transport and selectivity [38]. Recently, hypercrosslinked polymers (HCPs) with porous organic frameworks have become an important issue. HCPs usually exhibit free-defect interfaces in the course of fabrication of MMMs thanks to good compatibility between fillers and polymer matrices. More importantly, HCPs are a type of novel filler with fine prospects in MMMs due to their good ability to capture gas [39]. Some reports have indicated that use of HCPs for preparation of MMMs can break through the trade-off effect and greatly facilitate gas transport [40]. However, there are limited reports on the use of HCPs for preparation of HCPs/PI MMMs, and there are few studies on the gas transport of HCPs/PI MMMs. Distinctly explaining the gas transport behaviors in HCPs/PI MMMs will provide theoretical support to design and manufacture PI MMMs.

In this work, we first prepared 6FDA-based PI MMMs, using novel HCPs as organic fillers. We then used 2-phenylimidazole and α, α-dichloro-p-xylene as reaction monomers, and the HCPs were synthesized through direct polymerization. Poly(2-phenylimidazole-co-α, α-dichloro-p-xylene) is simultaneously a weak acid and a weak base due to its special chemical structure. HCPs and 6FDA-based PI were blended to prepare the HCPs/PI MMMs through a solution-casting process. The gas separation properties and molecular simulations of the HCPs/PI MMMs were researched through separation of pure CO_2_/CH_4_ and O_2_/N_2_.

## 2. Results and Discussion

### 2.1. SEM Images of HCPs and PI Membranes

In Figure 1a, XRD shows that the as-prepared hypercrosslinked polymers are amorphous, which is attributed to the free polymerization and disorder crosslinking reaction of α,α′-dichloro-p-xylene and 2-phenylimidazole, resulting in nondirectional distortion and growth of the spatial structure. The FT-IR spectrum in Figure 2b shows that the HCPs had four characteristic peaks, at 3402 cm^−1^, 2923 cm^−1^, 1603 cm^−1^, and 1437 cm^−1^, respectively. These four characteristic peaks are attributed to N-H in imidazole, C-H in -CH_2_-, C=C in the benzene ring, and C-N in benzylamine, respectively, indicating that the 2-phenylimidazole-type hypercrosslinked polymers were successfully synthesized. The BET adsorption and desorption curve is type II, and the adsorption and desorption curves do not overlap due to the existence of a large number of pore structures. The specific surface area of the HCPs was 721.37 m^2^/g, and the pore size distribution of the HCPs, calculated with BJH, was approximately 5–10 Å, which proves that pore size of HCPs is mainly microporous, as in Figure 1c. In Figure 1c (insert), the particle size distribution shows that the particle size of the HCPs is about 102.02 nm. In Figure 1d, SEM shows that the HCPs are irregular spheres.

Whether HCPs can be uniformly dispersed in MMMs is an important indicator for evaluating the quality of MMMs. Gas separation performance will greatly reduce once HCPs agglomerate. Figure 2a,b show the surface SEM image and the cross-sectional SEM image of pure PI film, respectively. A pure PI film with a homogenous structure is relatively smooth and dense. Figure 1e and Figure 2c,g,j show the surface SEM images of the HCPs-0.02/PI MMMs, the HCPs-0.04/PI MMMs, the HCPs-0.06/PI MMMs, and the HCPs-0.08/PI MMMs, respectively. The results showed that the amount of HCPs in MMMs increases with continued addition of HCPs. When the amount of HCPs is 0.08 wt.%, the size of the HCPs becomes relatively larger because slight agglomeration occurs due to these additional HCPs. Fortunately, HCPs have no large-scale agglomeration and are still dispersed well in PI film, indicating good compatibility between HCPs and PI film [29]. 

Figure 2d,f,h,j show cross-sectional SEM images of the HCPs-0.02/PI MMMs, the HCPs-0.04/PI MMMs, the HCPs-0.06/PI MMMs, and the HCPs-0.08/PI MMMs, respectively. The distribution and sizes of the HCPs agreed with the SEM images. All SEM images show that the as-prepared HCPs/PI MMMs had no defects after the HCPs were added, avoiding gas leakage during gas separation and providing a better guarantee for the next step of gas separation.

### 2.2. FT-IR Spectra of PI Membranes

The FT-IR spectra of pure PI film and different HCPs/PI MMMs are shown in Figure 3. The wavenumbers at 1785 cm^−1^ and 1725 cm^−1^ are the symmetric and asymmetric stretching vibration peaks of C=O in the imide groups, while the wavenumbers at 1370 cm^−1^ and 718 cm^−1^ are the stretching and bending vibration peaks of C-N [41,42]. In addition, the assigned peaks at 1250 cm^−1^ and 1140 cm^−1^ are the stretching vibrations of C-F in PI, demonstrating that fluorinated polyimides were successfully obtained [43]. The characteristic peaks of different HCPs/PI MMMs are the same as those of pure PI film, indicating that no new chemical bond was formed between the HCPs and the PI.

### 2.3. Mechanical Strength of PI Membranes

MMMs need to withstand certain pressures during gas separation. Therefore, MMMs should have sufficient mechanical strength. The mechanical strength of the HCPs/PI MMMs was necessarily assessed. The elongation at breaking and the tensile strength of pure PI film and different PI/HCPs MMMs are displayed in Figure 4. As shown in Figure 4, after the HCPs were added, the elongation at breaking of the HCPs/PI MMMs gradually decreased with increasing amounts of HCPs. However, the elongation at breaking of the HCPs/PI MMMs was higher than that of pure PI film. In addition, the values of the elongations at breaking of pure PI film and HCPs/PI MMMs were not large, and the value of the elongation at breaking of the HCPs/PI-0.02 was the highest (only 3.9%). This may be because PI has high rigidity and molecular chain motion is difficult. The tensile strength of the HCPs/PI MMMs increased in the beginning and then decreased. It reached up to a maximum of about 44.2 MPa when the amount of HCPs was 0.04 wt.%. HCPs are a type of pure organic filler. They have good compatibility with PI, and therefore, HCPs/PI MMMs have no defects, and their tensile strength does not decrease. In addition, the presence of 2-phenylimidazole in HCPs and hydrogen bonds between the N-H bond on the imidazole ring and the C=O bond in PI can increase the tensile strength of HCPs/PI MMMs due to incorporation of HCP fillers; the result is similar to those of previous reports in which filler enhanced the mechanical strength of MMMs [44,45]. The tensile strength of HCPs/PI MMMs reduces because of excessive HCPs that may be related to the interfacial effect forming agglomeration of HCPs.

### 2.4. Gas Separation Performance

Gas separation performance is an important ultimate index to evaluate the quality of MMMs. Figure 5a shows the gas separation performances of pure PI film and HCPs/PI MMMs for CO_2_/CH_4_ at 35 °C and 1 bar of feed gas pressure. The permeability of pure PI film toward CO_2_ was 62.41 Barrer, and the ideal selectivity to CO_2_/CH_4_ was 20.76. Simultaneously, as the amount of HCPs varied from 0.02 wt.% to 0.08 wt.%, the permeability of HCPs/PI MMMs to CO_2_ was 63.36 Barrer, then 105.85 Barrer, 86.22 Barrer, and 68.58 Barrer. It increased at first and then decreased. Moreover, the permeability of all HCPs/PI MMMs was higher than that of pure PI film. When the amount of HCPs was 0.04 wt.%, the permeability of HCPs/PI MMMs to CH_4_ and CO_2_ enhanced by 125% and 69.6%, respectively. The results demonstrate that HCPs/PI MMMs facilitate CH_4_ and CO_2_ transport. In addition, the permeability of HCPs/PI MMMs is better than that of pure PI film. The ideal selectivities are 14.57, 15.67, 14.66, and 13.59. The ideal selectivity of all HCPs/PI MMMs only decreases slightly from that of a pure PI film.

To better assess the gas transport properties of MMMs, the CO_2_ and CH_4_ permeability and ideal selectivity data are displayed in Figure 5b in a Robeson’s diagram for the corresponding CO_2_ and CH_4_ pair. The ideal selectivity obviously decreases, as Robeson defines the linear relationship as the “upper bound” accompanied by an improvement of gas permeability in MMMs. Here, CO_2_ and CH_4_ are below the upper-bound line and represent the perfect combination of permeability and ideal selectivity for the fixed gas pair. However, they show a favorable tendency to increase gas permeability, while the ideal selectivity basically remains unchanged after addition of HCPs, which is different from the typical trade-off phenomenon of polymer membranes. Obviously, HCPs can be used to prepare MMMs in an effective approach to successfully improve gas transport and break the trade-off effect.

Next, O_2_ and N_2_ permeability experiments were carried out to prove that HCPs can promote different gas transport. The O_2_/N_2_ separation performance of the PI film and different HCPs/PI MMMs and Robenson’s upper bound correlation (2008) [46] for O_2_/N_2_ separation are shown in Figure 6. Figure 6a shows the gas separation performance of pure PI film and HCPs/PI MMMs for O_2_/N_2_. According to Figure 6a, the pristine PI film exhibited that the permeability of O_2_ was 13.85 Barrer and the ideal selectivity of the O_2_/N_2_ pair was 3.29. Meanwhile, the permeability of HCPs/PI MMMs to O_2_ increased at first and then decreased as the amount of HCPs changed from 0.02 wt.% to 0.08 wt.%. Moreover, the permeability of the HCPs/PI MMMs to O_2_ was higher than that of pristine PI film. When the HCPs were 0.04 wt.%, the permeability of the HCPs/PI MMMs to O_2_ was 24.03 Barrer. Compared with the pristine PI film, the permeability of the HCPs/PI MMMs toward O_2_ and N_2_ increased by 73.5% and 91.6%, respectively, while the ideal selectivity of the HCPs/PI MMMs toward the O_2_/N_2_ pair only reduced by 9.7%. These results demonstrate that HCPs/PI MMMs facilitate O_2_ and N_2_ transport, improve gas permeability, and maintain stable selectivity.

The permeability and ideal selectivity results in the Robeson’s diagram for the corresponding O_2_ and N_2_ pair (Figure 6b) are analogous to those of the corresponding CO_2_ and CH_4_ pair; however, the permeability and ideal selectivity of the HCPs/PI MMMs to O_2_/N_2_ are obviously lower than those for CO_2_/CH_4_. HCPs with imidazole-type polyionic liquid structures have certain catalytic properties for the cyclization of CO_2_ to carbonate. These can be used as CO_2_ capture and conversion materials [47,48,49,50]. The interactions between different gases and MMMs probably facilitate CO_2_ and O_2_ permeability. In order to analyze different gas moving behaviors, molecular simulation was carried out as described in the following section.

### 2.5. Analysis of Gas Separation Process

The diffusion coefficients and sorption capacities of the CO_2_/CH_4_ and O_2_/N_2_ pairs are shown in Figure 7a–d. It can be seen from Figure 7a–d that the sorption capacity of CO_2_ (O_2_) in the HCPs/PI MMMs was weaker than that of CH_4_ (N_2_), while the diffusion of CO_2_ (O_2_) in the HCPs/PI MMMs was easier than that of CH_4_ (N_2_), indicating stronger interaction of CH_4_ (N_2_) and HCPs/PI MMMs. According to Equation4, the sorption capacities of different gases were 1.03 × 10^−3^ g/g (CO_2_), 0.91 × 10^−3^ g/g(CH_4_), 0.39 × 10^−3^ g/g (O_2_), and 0.98 × 10^−3^ g/g (N_2_), respectively, demonstrating that the sorption capacity of CO_2_ is rather higher than that of CH_4_ and other gases (O_2_ and N_2_). According to Equation (5), the permeability coefficients were 3.67 (CO_2_/CH_4_ pair) and 0.32 (O_2_/N_2_ pair), indicating that the ideal selectivity of the CO_2_/CH_4_ pair was more excellent than that of the O_2_/N_2_ pair. The distribution of binding energies of the HCPs, the PI, and the HCPs/PI is shown in Figure 7e. It can be seen that the binding energies of the HCPs, the PI, and the HCPs/PI are similar in peak shape, demonstrating that the compatibility of HCPs and PI is very good. In addition, the lowest energy conformation is displayed in Figure 6f. As shown in Figure 7f, an HCP molecule can be very close to a PI chain, further proving their perfect compatibility.

The gas separation performance of MMMs strongly depends on the MMM structures. To gain an insight into the relation between the permeability and selectivity of MMMs and microstructure, the average interspacing distance (d-spacing) stands for the distance between polymer chain segments that reflects the permeability coefficient of MMMs. The d-spacing value of MMMs can be calculated with Bragg’s equation [28]. XRD patterns of the PI film and the HCPs/PI MMMs are shown in Figure 8. The results demonstrate that the PI film and the HCPs/PI MMMs have broad, amorphous peaks that are attributed to the presence of bulky -C(CF_3_)_2_- groups in PI; thus, the PI film and HCPs/PI MMMs have loose chain packing [51].

Generally, the amorphous nature of PI is favorable to gas permeation. The gas permeability of a polymer membrane will strengthen with increased d-spacing. The d-spacing values for the PI film and the HCPs/PI MMMs were 5.58 Å, 5.60 Å, 5.70 Å, 5.70 Å, and 5.46 Å (Table 1). The results show that the d-spacing trend for the PI film and the HCPs/PI MMMs gradually increased and then reduced, which agrees with different gas permeability observations except for that of the HCPs-0.06/PI MMMs. Although the d-spacing value of the HCPs-0.04/PI MMMs is equal to that of the HCPs-0.06/PI MMMs, the gas permeability of the HCPs-0.04/PI MMMs was higher than that of the HCPs-0.06/PI MMMs. It is obvious that the free spaces in the microstructure beside the d-spacing of the polymer chain segments have a fundamental effect on the gas permeability of a polymer membrane [43].

Figure 9 shows the molecular simulation diagrams of the PI and HCPs/PI systems after MD simulation. In this simulation, Figure 9a–e are schematic representations of simulated molecular cells of the PI and HCPs/PI systems, in which twenty-three molecular chains composed of 6FDA-DAPI, with 30 repeating units for the PI, were placed in a cubic box. The blue and gray parts indicate free volume and occupied volume, respectively. Figure 9 shows that the free volume of the HCPs/PI MMMs was significantly larger than that of pure PI film. According to the literature, large free volume provides a fast channel for gas transport, thereby enhancing gas permeability [27,43].

The density of HCPs/PI mixed polymers in the stable state varied from 1.5358 to 1.3553 g/cm^3^, which is higher than that of 6FDA-based polyimides, and the FFV of MMMs (0.1892–0.2498) is also higher than that of most 6FDA-based polyimides [52]. Combining Figure 9 with Table 1, we can see that the FFVs of MMMs containing different HCP amounts are larger than that of pure PI film. As the amount of HCPs increased, the FFVs of the PI MMMs first increased and then slightly decreased. Therefore, although the permeability of HCPs/PI MMMs to different gases is higher than that of pure PI film, the gas transport of HCPs/PI MMMs does not always increase with more HCPs. When the amount of HCPs added was 0.04%, the FFVs of the HCPs/PI MMMs reached a maximum (0.2498). Therefore, the permeability of different gases was the largest. This permeability gradually reduced as even more was added. This also proves that gas permeability not only is related to the d-space of the membranes but also has a greater relationship with the FFVs of the membranes. In contrast, the ideal selectivities of different gas pairs reduce according to large FFVs; this result agrees with the trend of ideal selectivity in Figure 4a and Figure 5a.

## 3. Experimental Section

### 3.1. Materials and Chemicals

Hexafluoroisopropylidene diphthalic anhydride (6FDA, Tianjin Zhongtai Chemical Technology Co., Ltd., Tianjin, China), diaminophyenylindane (DAPI, Gladesland (Tianjin) Pharmaceutical Technology Co., Ltd., Tianjin, China), dimethylacetamide (DMAc, Tianjin Reagent Supply and Marketing Company, Tianjin, China), acetic anhydride (CH_3_CO)_2_O, Tianjin Bohai Chemical Reagent Co., Ltd., Tianjin, China), trimethylamine (TEA, Tianjin Reagent Supply and Marketing Company, Tianjin, China), 2-phenylimidazole (C_9_H_8_N_2_, Shanghai Aladdin Biochemical Technology Co., Ltd., Shanghai, China), α, α’-dichloro-p-xylene (C_8_H_10_, Shanghai Aladdin Biochemical Technology Co., Ltd., China), 1,2-dichloroethane (CH_2_Cl_2_, Tianjin Huihang Chemical Technology Co., Ltd., Tianjin, China), anhydrous ferric chloride (FeCl_3_, Shanghai Macklin Biochemical Co., Ltd., Shanghai, China), and methanol (CH_3_OH, Tianjin Damao Chemical Reagent Factory, Tianjin, China) were analytical-grade and used without further purification. Nitrogen (N_2_), methane (CH_4_), carbon dioxide (CO_2_), and oxygen (O_2_) were high-purity and purchased from Tianjin West Development Co., Ltd. Distilled water was made with a homemade distillation device.

### 3.2. Preparation of PI MMMs

#### 3.2.1. Synthesis of Hypercrosslinked Polymers (HCPs)

The synthetic route of HCPs refers to the literature [53,54]. The process is as follows: 5 mmol of 2-phenylimidazole and 5 mmol of α, α′-dichloro-p-xylene were fully dissolved in 10.00 mL of 1,2-dichloroethane, and then 10 mmol of FeCl_3_ was added into the above solution. The system was polymerized at 80 °C for 24 h under a nitrogen atmosphere. The resulting brown product was stirred in methanol for 24 h to remove unreacted monomers and anhydrous FeCl_3_. After the product was filtered and washed with ultrapure water, it was dried in a vacuum oven at 60 °C for 12 h. The final product was denoted as HCPs. The synthetic process is shown in Figure 1.

#### 3.2.2. Synthesis of 6FDA-Based PI

In a typical experiment, 0.015 moL of DAPI and 0.015 moL of 6FDA were added into 46.25 mL of DMAc solution in a three-neck flask and fully dissolved with mechanical stirring, and the polymerization system was performed at 0 °C for 24 h. Then, 0.045 moL of acetic anhydride as a dehydrating agent and 0.015 moL of TEA as a catalyst were doped into the reactive system, and the reactive temperature was raised to 30 °C. The polymerization system further polymerized for another 24 h. The resulting product was precipitated with anhydrous methanol. Finally, it was dried in a vacuum oven at 170 °C for 24 h. The white powder obtained was named as 6FDA-based PI. This process is according to Freeman [55], and the synthesis process of 6FDA-based PI is shown in Figure 2. 

#### 3.2.3. Preparation of PI MMMs

The HCPs were added to 5.0 mL of DMAc solution and stirred for 2 h (the different mass ratios of the HCPs and the PI were 0.02 wt.%, 0.04 wt.%, 0.06 wt.%, and 0.08 wt.%, respectively). We then continued dispersing for another 1 h via ultrasonic treatment after 0.500 g of PI was added into the above system, and we stirred for 12 h to totally dissolve. The casting solution was evenly coated on an ultraflat watch glass, and the DMAc was evaporated at 60 °C. Finally, the formed membrane was laid on a common watch glass for heat treatment at 60 °C for 2 h, 120 °C for 4 h, 180 °C for 4 h, and 220 °C for 12 h. The PI MMMs were obtained and labeled as HCPs-0.02/PI, HCPs-0.04/PI, HCPs-0.06/PI, and HCPs-0.08/PI. (A blank experiment with pure PI film was prepared via the steps aforementioned, except for the addition of HCPs.)

### 3.3. Characterization

The crystalline phases of the HCPs and the HCPs/PI MMMs was determined with an X-ray diffractometer (XRD, Bruker D8 Discover, Salbruken, German). After the powders were ground with an agate mortar, they were put on a glass slide with a circle groove and tested at 10°/min sweep speed with a Cu target. The tube voltage, the current, and the scan range were 40 kV, 200 mA, and 5–50°, respectively. The chemical structures of the HCPs and HCPs/PI MMMs were characterized through Fourier transform infrared spectroscopy (FT-IR, Bruker Vertex 70, Salbruken, German). The test range of the HCPs was 4000–600 cm^−1^, tested with a KBr tablet method. The test range of the MMMs was 4000–600 cm^−1^ via a reflection mode test. The morphologies of the HCPs and the HCPs/PI MMMs adhered to the carbon-conductive adhesive and were sprayed with gold and detected with a scanning electron microscope (SEM, Hitachi S4800,Tokyo, Japan). N_2_ adsorption was used to measure the specific surface areas and pore size distributions of the HCPs (BET and BJH, micromeritics ASAP2460, Norcross, GA, USA). The mechanical strengths of the HCPs/PI MMMs were tested using a universal stretching machine (XD-121A, Shanghai Xinda Instrument Co., Ltd., Shanghai, China). The tested membranes were cut into 6 cm × 1 cm strips to test their tensile strength and elongation at breaking.

### 3.4. Evaluation of Gas Separation Performance

Single gas-permeation performances of pure PI film and HCPs/PI MMMs were measured with the variable-volume method. After these membranes were placed into a porous supporter, the system was sealed in a specific membrane chamber. The feed gas pressure was conveniently adjusted with a suction pressure gauge. Gas permeability was determined in a sequence of CH_4_, N_2_, O_2_, and CO_2_ at 308.15 K and under 1 bar, with an effective membrane area of approximately π cm^2^. The frame cell was degassed for 1 h to remove residual air. The pressure on the lower side of the membrane chamber was controlled at −1 bar. The experimental data were recorded when the system was kept at a stable state.

Permeability (*P*, Barrer) was the averaged value from thrice-repeated experiments. According to the growing rate of downstream pressure with time in the steady state, *P* was calculated with Equation (1), as below:(1)$$P=\frac{273.15}{76\times T}\times \frac{Vl}{\Delta P\times A}\frac{d_{p}}{d_{t}}$$

Here, *V* is the gas volume under the membrane, *A* is the effective membrane area (cm^2^), *T* is the experimental temperature (*K*), l is the thickness of the membrane (cm), and ∆*P* is the transmembrane pressure difference (cmHg), while *d_p_*/*d_t_* (cmHg/s) is the rate of pressure increase at the permeate side in the steady state.

The ideal selectivity for two gases is calculated via Equation (2):*α_A_*_/*B*_ = *P_A_*/*P_B_*(2)

Here, *P_A_* is the gas permeability of pure gas *A*, and *P_B_* is the gas permeability of pure gas *B*.

### 3.5. Molecular Simulations

Molecular modeling and computation were conducted using Materials Studio. The universal forcefield was used in the calculations of diffusion coefficient, sorption capacity and binding energy. The geometry optimization in the Forcite module was used to relax both the PI_55_ and the HCP with a smart algorithm. An amorphous cell box, P1, comprising 1 PI_55_ and 6 HCP, was constructed. For the simulation of the CH_4_ and CO_2_ system, 1 CH_4_ and 10 CO_2_ were added into box P1, while for the simulation of the N_2_ and O_2_ system, 1 N_2_ and 4 O_2_ were added. Geometry optimization was used to optimize the box with a fine quality. Then, a simulated annealing procedure with 8 cycles was applied to the systems. The initial and midcycle temperatures were 300 K and 500 K, respectively. The systems were simulated for 30 ps in an NVT (constant volume and temperature) ensemble. Finally, the optimization box was 90.3 × 94.3 × 92.0 Å, with a density of 1.014 g/cm^3^. The molecular dynamics were calculated in a constant-energy, constant-volume ensemble (NVE) for 30 ps. We calculated the diffusion coefficient (D) using molecular mean-square displacement (MSD). The *D* values of CO_2_, CH_4_, O_2_, and N_2_ in HCPs/PI MMMs could be calculated with Equation (3):(3)$$D=\frac{1}{6N_{\alpha }}\underset{t\to \infty }{\lim}\frac{d}{dt}\sum_{i=1}^{N_{\alpha }}\langle {\left[r_{i}\left(t\right)-r_{i}\left(0\right)\right]}^{2}\rangle$$
where *N*_α_ is the number of diffusive atoms in the system.

The sorption module was used to calculate a loading curve for different gas molecules in a periodic box, P1. The adsorption isotherm was calculated using the Metropolis method with a fine quality. Both CO_2_ and CH_4_ were set to be sorbates. The lower and upper bounds of fugacity were set to be 50.6625 kPa and 101.325 kPa, respectively. The temperature was set to 298 K. The same settings were also used in the adsorption isotherm of O_2_ and N_2_.

The interactions between PI_55_ and HCP were investigated in the blend module, with a fine quality. PI_55_ and HCP were set to be the base and screen components, respectively. Both the energy and cluster samples were set to be 10,000. The energy bin width was set to be 0.02 kcal/mol. The iterations per cluster were set to be 20. The sorption capacity (*S*) and permeability coefficients (*P_c_*) could be calculated using Equations (4) and (5):(4)$$S=\frac{m_{g}}{m_{p}}=\frac{n\times M}{\rho \times V\times N_{A}}$$
(5)$$P_{c}=\frac{D_{a}\times S_{a}}{D_{b}\times S_{b}}$$
where m_g_ is the gas molecular mass, m_p_ is the mass PI_55_ and HCPs in an optimization box, n is the number of gas molecules, M is the molar mass of the gas molecules, *ρ* is the density of the optimization box (1.014 g/cm^3^), *V* is the volume of the optimization box (90.3 × 94.3 × 92.0 Å), *N*_A_ is Avogadro’s constant (6.02 × 10^23^), *D_a_* and *D_b_* are the diffusion coefficients of different gases in MMMs, and *S_a_* and *S_b_* are the sorption capacities of MMMs for different gases.

The FFVs of the PI film and the HCPs/PI MMMs were calculated with molecular dynamic (MD) simulation. The Gromacs 2018.4 package was used to perform all molecular models for the MD simulation. In detail, twenty-three molecular chains composed of dianhydride and diamine, with 30 repeating units (including a large HCPs unit in the MMMs) for each, were placed in a cubic simulation cell, and the best use of the balance between cell dimensions and calculation time was made. Energy minimization processes of over 800 iterations, to obtain stable structures, were used to perform all of the molecular models. Then, the MD calculations were carried out under an NVT ensemble for a duration of 100 ps at 300 K. Newton’s second law of motion was applied to estimate the dynamic behaviors of polymer molecules. The theoretical calculations adopted the COMPASS forcefield. The FFVs of the PI film and the HCPs/PI MMMs could be calculated using Equation (6):*FFV* = (*V*_0_ − *V*_1_)/*V*_0_ = (*V* − 1.3*V*_2_)/*V*_0_(6)

Here, *V*_0_, *V*_1_, and *V*_2_ are the specific volume, the occupied volume, and the van der Waals volume of 6FDA-based PI, respectively.

## 4. Conclusions

In this work, HCPs/PI MMMs were successfully obtained, with HCPs containing 2-phenylimidazole structure as a filler and 6-FDA-based PI as a polymer matrix. The as-prepared MMMs were characterized with XRD, FT-IR, SEM, and a universal tensile testing machine. The results show that the as-prepared HCPs/PI MMMs were amorphous, with no chemical bonding between HCPs and PI. The HCPs could be easily dispersed in the PI matrix. Even at 0.08 wt.% HCP, the MMMs still did not cause defects at the interface between HCPs and PI due to their perfect compatibility other than slight agglomeration. 

The MMMs had good mechanical properties due to the structure of PI itself and the presence of HCPs. When the HCPs were 0.04 wt.%, the tensile strength at the break of the MMMs was as high as 44.2 MPa, which is much greater than the 24.0 MPa of pure PI film. Gas permeation experiments showed that the MMMs had better gas transport capabilities than did pure PI film, especially for CO_2_; the permeability was improved, and the ideal selectivity basically remained unchanged. The permeabilities of CO_2_ and O_2_ were 69.6% and 73.5% higher than that of pure PI film, respectively, and the permeabilities of CH_4_ and N_2_ were 124.0% and 91.6% higher than that of pure PI film, respectively. However, the ideal selectivities of the MMMs for the CO_2_/CH_4_ and O_2_/N_2_ gas pairs were only reduced by about 24.5% and 9.7%, respectively. The molecular dynamics simulations showed that the permeability trend of MMMs agrees with their FFV changes. The HCPs used to fabricate MMMs have potential applications in the fields of natural gas purification and O_2_ enrichment.

## Data Availability

Data sharing not applicable.

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
