# Peer review of "Fabrication, Facilitating Gas Permeability, and Molecular Simulations of Porous Hypercrosslinked Polymers Embedding 6FDA-Based Polyimide Mixed-Matrix Membranes"

_molecules, 2023, doi:10.3390/molecules28052028_

Round 1

Reviewer 1 Report

The manuscript deals with preparation, characterization and separation performance of mixed matrix membranes based on PI (6-FDA) matrix and HCPs (2-phenylimidazole) fillers. MMMS are compared with a pristine PI membrane. The prospective application of this new membrane materials is stated to be natural gas purification and O2 enrichment.

The presented results are interesting and materials developed were studied in a comprehensive way. However, I have a few suggestions and remarks in order to improve the general quality of the manuscript. I would like to see revised version.

1) I recommend to review carrefully the English style. The text is unclear in several places or seems to be grammatically or stylistically incorrect. Examples:

- ... they facilitate gas transport of fillers, including plenty of porous structures and functional groups (p.3),

- More significantly, the largely fractional free volume (FFV) ... (p.3)

- researchers have focused on the utility of fillers (p.4)

- the incorporation of ZMOF filler with an affinity CO2 framework (p.4)

- Clearly explaining the gas transport in HCPs/PI MMMs has an important impact on the structural design and manufacturing process of PI MMMs (p.5)

- The polymerization continued polymerizing for another 24 h (p.7)

- Permeability (P, Barrer) was the averaged value from three times experimental results (p.10)

- Besides, HCPs/PI MMMs have better transport than pure PI film (p.18)

- the permeability of the HCPs/PI MMMs to O2 is 24.03 Barrer and plateaus. (p.19-20)

- molecular simulation has to been carried out (p.20)

- blends binding energy distribution (p.21)

- ideal selectivity is maintained without drastically decreasing (p.26)

2) The following statements require reconsideration:

- “Unfortunately, the permeation has always been a problem restricting gas separation performance in membrane technology” (p2.): the term permeability instead of permeation will sounds better in this context.

- “Here, the CO2 and CH4 are below the upper-bound line and represent the perfect combination of permeability and ideal selectivity for the fixed gas pair. However, they show a favorable tendency to increase the gas permeability, while the ideal selectivity basically remains unchanged after adding HCPs, which is different from the typical trade-off phenomenon of polymer membranes.”: Why this is “the perfect combination”. This is not strictly true that “ideal selectivity basically remains unchanged after adding HCPs”. While comparing HCPs-0.04/PI with the pristine PI the trade-off is similar to the Robeson line (connect, for example, blue square with red triangle in Fig.4). This is the best case.

3) “Hexafluoroisopropylidene diphthalic anhydride (6FDA), diaminophyenylindane (DAPI), dimethylacetamide (DMAc), acetic anhydride, trimethylamine (TEA), 2-phenylimidazole, α, α´-dichloro-p-xylene, 1,2-dichloroethane, anhydrous ferric chloride (FeCl3), and methanol were analytical grade and used without further purification.”: mention a supplier of these substances.

4) In the section 2.5 Molecular simulations try to describe simulations and their assumption in a way other than citing a sofware manual. Avoid using sentences like “Adsorption isotherm as the Task and Metropolis for the Method.” “Set the Quality to fine.”Explain abreviations, like MSD. Indexes like for PI55.

5) In my opinion Figs 6 c-d present sorption capacity (number of molecules per cell) not sorption energy. Moreover, I would expect that the sorption capacity of CO2 should be rather higher than that of CH4 and other gases in this study. Having these theoretical “diffusivity” and “sorption capacity” one is able to estimate permeability coefficients of gas in the membrane (or at least ideal selectivity) and compare them to the experimental values presented earlier in this study. Include, please such a discussion. I appreciate this theoretical considearation and like it but it require in my opinion a kind of experimental validation.

6) In table 1 the density is constantly decreasing with an increase in HCPs content despite the increase in occupied volume. This could be explainable with a growth of FFV but the latter drops starting from HCPs-0.06 /PI. Try to explain why.

7) “It is very possible that the free spaces in the microstructure besides the d-spacing of polymer chain segments have a fundamental effect on the gas permeability of polymer membranes[46]. (p.23) ” This is obvious statement and in fact you are generating MMMs to have higher FFV in order to improve permeability. Similar statement is on the next page “According to the literature, the large free volume provides a fast channel for gas transport, thereby enhancing gas permeability.”

8) “In this simulation, Fig. 8a and 8b are schematic representations …” (p.24). Only Figs 8a and 8b?

9) “Fig. 8 shows that the free volume of HCPs/PI MMMs is significantly larger than that of pure PI films.” Taking into account this figure HCPs-0.04 /PI (with the largest FFV) seems to have FFV similar to pristine PI and lower than HCPs-0.02/PI and HCPs-0.08/PI.

10) “In contrast, the ideal selectivity of different gas pairs reduces according to the large FFV; this result agrees with Fig. 4 and Fig. 5.” Where in this theoretical section is shown an ideal selectivity to be compared in with experimental results in Figs 4 and 5.? (see  remark No 5).

Author Response

Dear Reviewer,

Thank you for your important and valuable comments. According to your suggestion from answer 1 to answer 10, we revised our manuscript on every point and marked in blue fronts. Please download the coverletter to look at the reply.

Warmly regards, 

Yinhui Li

Reviewer 2 Report

Mixed matrix membranes (MMMs) with heterogeneous structures are excellent candidates for gas separation. The largely
fractional free volume (FFV) of aromatically fluorinated PI used by autors,  is beneficial to gas transport but
large FFV will result in gas selectivity reductio that falls into conventional trade-off effects. Hyper-crosslinked polymers (HCPs) with porous organic frameworks have become an interesting issue. HCPs usually exhibit a free-defect interface in the course of fabrication of MMMs
thanks to good compatibility between the pure organic fillers and polymer matrix. More importantly, HCPs are a kind of novel prospectif  filler  due to their good ability to capture gas.

Unfortunately, the synthesized HC polymer is not characterized in any way in terms of its specific surface. It is necessary to do nitrogen adsorption (BET) and to fully assess its porosity and adsorption CO2. The mass fraction of the polymer in MMM is not large, and the influence on the gas transport properties of the material is very noticeable. For a more complete understanding of the effects found, the characterization of 6F polymer by the BET method would also be useful (it is possible that its specific surface was studied in Freeman's works). The absence of the above characteristics reduces the scientific value of the study (but not in itself almost an important result). The article can be published when additional data is submitted.

Author Response

Dear reviewer,

Thank you for your important comment. Accoding to your suggestion, the BET result is added into the text. Besides, we added XRD, FT-IR and SEM characterization. The suggestion is very valuable to this manuscript.

Warmly regards,

Yinhui Li
